# Ultrafast Photo-Ion Probing of the Relaxation Dynamics in 2-Thiouracil

**DOI:** 10.3390/molecules28052354

**Published:** 2023-03-03

**Authors:** Matthew Scott Robinson, Mario Niebuhr, Markus Gühr

**Affiliations:** Institut für Physik und Astronomie, Universität Potsdam, Karl-Liebknecht-Straße 24/25, 14476 Potsdam, Germany

**Keywords:** thiouracil, uracil, pump-probe spectroscopy, ultrafast mass spectroscopy, PEPICO, ultrafast dynamics, ultrafast relaxation processes

## Abstract

In this work, we investigate the relaxation processes of 2-thiouracil after UV photoexcitation to the S_2_ state through the use of ultrafast, single-colour, pump-probe UV/UV spectroscopy. We place focus on investigating the appearance and subsequent decay signals of ionized fragments. We complement this with VUV-induced dissociative photoionisation studies collected at a synchrotron, allowing us to better understand and assign the ionisation channels involved in the appearance of the fragments. We find that all fragments appear when single photons with energy > 11 eV are used in the VUV experiments and hence appear through 3+ photon-order processes when 266 nm light is used. We also observe three major decays for the fragment ions: a sub-autocorrelation decay (i.e., sub-370 fs), a secondary ultrafast decay on the order of 300–400 fs, and a long decay on the order of 220 to 400 ps (all fragment dependent). These decays agree well with the previously established S_2_ → S_1_ → Triplet → Ground decay process. Results from the VUV study also suggest that some of the fragments may be created by dynamics occurring in the excited cationic state.

## 1. Introduction

Despite readily absorbing UV light, DNA and RNA nucleobases have the ability to undergo rapid relaxations through singlet and triplet manifolds to dissipate electronic energy into vibrational states [1,2,3,4,5,6,7]. This relaxation determines the remarkable photostability of the nucleobases and, in addition, is thought to be one of the important mechanisms contributing to the photostability of the genetic code by reducing the occurrence of UV-induced lesions [7,8,9,10,11,12].

Among the important UV-induced damages in nucleic acids, the cyclopyrimidine dimer (CPD) occurring among neighbouring thymine bases is the most abundant [13], and thus the UV-induced relaxation in thymine, as well as other nucleobases such as uracil, has been studied extensively [7,10,14,15]. However, naturally occurring variants of canonical nucleobases, similar to the commonly-studied thionated nucleobases, where one or more of the oxygen atoms is substituted for a sulphur atom, can appear with their own unique chemical features. The most striking of these is the efficient creation of long-lived triplet states [16,17,18], which pose a risk to live tissue due to the creation of reactive singlet oxygen as well as initiating interstrand crosslinking in DNA [9,19,20,21,22]. The suspected origin of this feature among thionucleobases is attributed to a lowering of the triplet excited state minima with respect to its conical intersections with the ground state. This barrier shift changes the likelihood that certain relaxation pathways will be explored compared to their canonical counterpart [23]. Already, a number of ultrafast relaxation studies of thionated nucleobases have been performed, providing us with opportunities to better understand the decay mechanisms of canonical and thionated nucleobases alike. [14,16,17].

One of the most studied thionucleobases is 2-thiouracil (2TU) [16,23,24,25,26]. For 2TU, both quantum calculations [26,27,28] and ultrafast experiments [2,24,29,30] indicate it will generally be excited by UV light to the S_2_ state (peaking at 4.2 eV/295 nm, ^1^π_S_π_6_^*^ in the orbital naming conventions suggested by Mai et al. [27]). A rapid internal conversion from the photoexcited S_2_ to the optically dark S_1_(^1^n_S_π_2_^*^) states is generally established as a doorway mechanism to lower triplet states. Pollum and Crespo-Hernández investigated the molecular dynamics in solution and reported a sub-200 fs decay for the S_2_ → S_1_ internal conversion [30], whilst computational work by Mai et al. [23,26,27], along with gas-phase experimental work by the Ullrich group [24,29,31,32] attribute a sub-100 fs decay to the internal conversion. After relaxation to the S_1_ state, intersystem crossing to the triplet manifold occurs after some hundred femtoseconds. In the solution phase, this process is attributed to an experimentally observed decay of 300–400 fs after a 316 nm excitation [30]. In the gas phase, the time constant is found to vary between 200 and 775 fs with increasing excitation wavelengths between 207 and 292 nm [31]. The molecule remains in these states for several ten-to-hundred picoseconds depending on the excitation wavelength [2,24,29,31]. Ultrafast X-ray Auger probing performed in our group allowed us to confirm the initial elongation of the C-S bond upon photoexcitation [33]. In addition, we performed ultrafast X-ray photoelectron spectroscopy on UV-excited 2TU, generally confirming the established pathways, but finding that part of the photoexcited population relaxes immediately into the ground state within 250 fs, as well as noting coherent 250 fs oscillations stemming from modulated population exchange between the S_1_ state and other states [34].

In this work, we look to further our understanding of the ultrafast dynamics of 2TU through lab-obtained UV/UV pump-probe photo-ion studies. Similar work by Ghafur et al. has been performed in the past [2], which was able to successfully elucidate some excited-state dynamics of 2TU. The work presented here advances the idea through unique comparisons to complementary vacuum ultraviolet (VUV) dissociative photoionisation data obtained at a synchrotron. These novel comparisons allow us to obtain an overall deeper understanding of the likely origins of each fragment ion, and how these appearances depend on the UV-induced dynamics in the pump-probe spectroscopy. These results help shape our picture of 2TU in terms of potential energy surfaces and molecular geometry changes in the relaxation process.

The remainder of this article is as follows: In Section 2 we present results, split into several subsections, detailing results from different UV/UV and VUV experiments. In Section 3, we interpret these results, discussing how they reflect on previously published work and improve our understanding of the relaxation processes of 2TU. In Section 4, we describe the experimental set-up and parameters used in both the ultrafast UV/UV ion fragmentation studies and the VUV studies performed at the Swiss Light Source (SLS). Finally, in Section 5 we present our conclusions.

## 2. Results

In this section, we present results from both the VUV and UV/UV experiments. We start by detailing what fragment ions are observed in both experiments (Section 2.1). This is then followed by results from the VUV work (Section 2.2) and UV/UV experiments (Section 2.3, Section 2.4 and Section 2.5), respectively. With respect to the Subsections that concentrate on the UV/UV experiments, whilst a general description of the setup can be found in Section 4.1, we briefly provide at the start of these Subsections the unique experimental parameters that are important to each of the UV/UV experiments discussed to help provide context to the reader.

### 2.1. Observed Fragments

We start with a brief summary of the highest abundancy fragments observed after the photoionisation of 2TU. An example from the UV/UV study can be seen in Figure 1, however, we mostly observe the same fragments in both the VUV and UV/UV experiments. Masses observed in both experiments include the 28, 41, 42, 58, 60, 69, 70, 95, and 100 atomic mass unit (amu) fragment ions as well as the parent (128 amu) ion. These appearances agree well with photoionisation mass spectra presented in past experiments [2,29,35,36]. Despite being observed, we will not discuss further the 58 and 60 amu fragments, as their signals were significantly weaker than other fragments. We will also not explicitly discuss the 70 amu fragment, as this always appeared as a shoulder to the 69 amu fragment ion signal, and has been previously suggested to be the same fragment as the 69, albeit with an additional hydrogen atom attached [37]. We note that only a weak signal was observed for the 96 amu fragment in the VUV studies to be presented below, and not at all in the UV/UV studies (most likely due to signal strength), and therefore, a discussion on this fragment will be limited. Similarly, due to its weak signal strength, discussion of the time-dependent features of the 100 amu ion is not possible, however, its appearance in the first place is interesting in the context of past uracil studies [1,2,38], and so a discussion based on this can be found in the Supplementary Data.

### 2.2. VUV (PEPICO) Studies

Figure 2 presents data from the VUV photoelectron-photoion coincidence (PEPICO) experiment performed at SLS. Data were collected using photons with energy (hv) in the range of hv = 8.55–15 eV, and is stitched together from three runs (see Section 4.2 for full details). As the main focus of this paper is the UV/UV experiment, only a coarse analysis of the VUV data will be made here to assist this work. Figure 2a shows a false-colour plot, on a logarithmic scale, detailing which mass fragments are observed in coincidence to the arrival of near-zero kinetic energy electrons at our detector for specific VUV photons.

Figure 2b shows the mass-specific lineouts taken from Figure 2a, which give the mass-selected threshold photoelectron spectra (ms-TPES) [39]. These show how the fragment-specific signals vary with respect to the VUV photon energy used. Whilst it is possible to draw analogies between an ms-TPES and photoelectron spectra, it is important to note that the two methods provide different information. Photoelectron spectra generally represent a full kinetic energy range, whilst ms-TPES represents a spectrum based on a narrow selective kinetic energy window (*ca*. 10 meV) in line with triggers from coincident ions. Nonetheless, ms-TPES is useful for the identification of ionisation thresholds with new cationic states, and how these states cause different fragmentation events.

In Table 1, we present the appearance energies of the ions of interest, fitted using the Asher model [40]. Whilst alternative fitting models that take into account a wider range of parameters for more accurate models are available (e.g., Rice–Ramsperger–Kassel–Marcus (RRKM) theory and Simplified Statistical Adiabatic Channel Model (SSACM) [41,42]) only approximate energies are needed to support the UV/UV work presented here, and hence the Asher model suffices.

### 2.3. UV/UV Power Series

Here, we discuss a power series scan performed using the UV/UV set-up to determine the photon dependency for each of the fragments of interest. This experiment is similar to the power-dependency studies of rare gases performed by L’Huillier et al. [43]. The scans were performed over a range of pump-probe delays, allowing us to identify time-dependent changes in the observed photon-dependencies (similar to work performed by Koch et al. [44]). Mass spectra were collected for a set of pump-probe pulse energies; a total of between 1.6 and 19.7 nJ was split evenly between the two beams, over pump-probe delays of 0, 0.1, 0.5, 1, 5, and 10 ps.

For each data point collected, the area under each mass peak of interest was integrated (referred to as the “Signal” from here on) and plotted against the combined pump-probe energy as a function of Log_10_(Signal) Vs Log_10_(Laser Energy). A selection of prominent mass peaks can be seen in Figure 3, whilst full plots for all ions of interest can be found in Appendix A in the Supplementary Data. In the region where the signal was both above the background and not saturating, the plots were fitted to a straight line to obtain a gradient; specifically, the parent ion, 69 amu fragment ion, and the remaining fragment ions of interest were fitted in the ranges of 1.6–9.3 nJ, 3.6–13.7 nJ, and 9.3–19.7 nJ, respectively. The values of the fitted gradients provide a strong indication of the number of photons needed to create each ion [43,44]. A summary of the fragment- and delay-dependent gradients can be found in Table 2.

As Table 2 details, the value of the fitted gradient for the parent ion is relatively constant at all pump-probe delays, with an average value of 1.9 ± 0.1. This is a strong indication that the parent ion is produced through a two-photon process when using 266 nm photons.

All of the fragment ions have slope values that are higher than 2, suggesting higher-order processes. In particular, values of the slopes for the 41, 42, 69, and 95 amu fragments all lie between 2.5 and 3 for all time delays measured, suggesting that three photons of 266 nm light are needed to produce these fragments. For the 28 amu fragment, we observe a slope of 2.9 at time zero, which then increases to between 3.2 and 3.9 for all other delays. This suggests a change in photon-order from 3 → 4 as one moves away from the pump-probe temporal overlap.

### 2.4. Time-Dependent UV/UV Signal from Pump-Probe Beams with Equal Power

The time-dependent photo-ion signals for the masses of interest when using a UV/UV pump and probe beams of equal powers were collected over three experiments. The first was a “low-power” scan, using a combined (i.e., incoherent summed) pulse energy of (7.4 ± 0.1) nJ, over a delay range of −1 → 1 ps; this allowed for a non-saturated parent ion signal to be collected. The second experiment was a “high-power” scan, using a combined pulse energy of 19.7 ± 0.1 nJ, over a delay range of −3 → 15 ps; this allowed for time-dependent data of the fragment ions around time zero region to be acquired. The third experiment was also a “high power” experiment (i.e., same laser settings), scanned over the range of 0 → 100 ps; this allowed for the longer decay processes of the fragments to be studied.

Figure 4 shows the time-dependent signals of the 28 amu (a), 69 amu (b), and parent ions (c). Plots for other ions can be found in the supplementary data in Appendix A. For all ions, a “fast” signal drop within the first picosecond around the time-zero position was observed. After this, the parent ion signal simply decayed to the background level. Most fragment ions show slow decays that remained above the background level for hundreds of picoseconds. Subplots have been included in Figure 4a,b to show the longer-lived decays for the 28 and 69 amu ions, respectively.

Fits of the data seen in Figure 4 were performed using a series of Gaussians convoluted with decaying exponential functions to determine the decay constants of the individual ions [45,46]. These fits exclude the coherent artefact that is seen around the time-zero delay for a number of ions. In a previous experiment, using a similar UV/UV set-up, we showed that these coherent artefacts arise due to the under-sampling of an ultrafast decay whose lifetime is shorter than the auto-correlation signal produced by chirped pulses used in the pump-probe experiment (as we have here; see Section 4.1 for details). We, therefore, infer that these artefacts similarly are the result of under-sampled, ultrafast decays, which we will discuss and assign in the due course of the paper. For more information on the origin of these artefacts, we direct readers to the Discussion and Supporting Information of our previous publication for full simulations [47]. The parent ion was initially fit to a Gaussian curve with an FWHM that matched the FWHM of the two-photon 1,4-dioxane ionisation signal observed in the auto-correlation experiment described in Section 4.1, convoluted with a single decaying exponential with a decay constant of τ_1_. However, τ_1_ simply tended to zero in the fitting process. This suggests that the shape of the parent ion signal closely matches the FWHM of the autocorrelation experiment.

For the 41, 42, 69, and 95 amu fragments, the fitting of the data was performed in two parts. First, to obtain a measure of the slow decay, a simple exponential decay function, with decay constant τ_3_, was fit to the data in the range of 1 to 100 ps. Next, fits of the data in the range of −3 to 15 ps were performed, using a Gaussian, whose FWHM matched the FWHM of the three-photon xenon signal collected in the auto-correlation experiments described in Section 4.1, convoluted with three decaying exponentials. During the fitting routine, the amplitudes and decay constants of two of the decaying exponentials, τ_1_, and τ_2_ were allowed to refine freely (after an appropriate initial estimate). The decay constant of the third exponential was fixed to the previously determined τ_3_ value, whilst its amplitude was allowed to refine freely. For all fragments, both the time-zero position and the FWHM of the Gaussian were refined for best fit, however, it was found that the FWHM never changed by more than a few femtoseconds from the values predicted by the auto-correlation experiments.

In performing the fits for the 41, 42, 69, and 95 amu fragments, it was found that the value of τ_1_ tended to zero, suggesting that the data around time zero was primarily the same as the auto-correlation signal. Whilst for the 41, 69, and 95 amu fragment ions it was possible to refine a unique value for τ_2_, in the case of the 42 amu fragment ion, the τ_2_ value either tended to the same near-zero value of τ_1_, or its amplitude tended to zero, suggesting that no unique τ_2_ value could be fit.

The fitting of the 28 amu fragment ion generally followed a similar procedure as with the other fragment ions. However, due to the four-photon dependency of the 28 amu fragment ion outside of the time-zero position, it was decided that a 300 fs FWHM Gaussian should be used to represent the auto-correlation in the convolution to fit the data, better matching the expected duration of a four-photon auto-correlation signal of our set-up. A summary of the fitted decay constants can be found in Table 3.

### 2.5. Time-Dependent UV/UV Signals from Pump and Probe Beam with Unequal Powers

In these experiments, variable ND filters were inserted into the Michelson Interferometer, allowing for unequal pump and probe powers. The probe beam was fixed to a pulse energy of 3 nJ, whilst the pump beam was scanned over a set of pre-selected pulse energies of 1.5, 3, 6, and 9 nJ. The insertion of these ND filters into the beam line caused the pump-probe temporal overlap to be shifted slightly from that used in Section 2.3 and Section 2.4. due to slight differences in the thickness of the ND filters (inherent in their manufacturing). Time-dependent scans were performed over the region ±1.5 ps around the time-zero position located in the previous experiments, and the data presented here has been adjusted so as to place the time-zero position at the point of the most intense signal.

Figure 5 shows the time-dependent signals for the parent and 69 amu fragment ion using pump and probe beams of unequal power. Plots relating to the other fragment ions are presented in Appendix A of the Supplementary Data. In this figure, we plot both the pump-probe data (i.e., positive delay times where the pump beam arrives first) and the probe-pump data (i.e., negative delay times where the pump beam arrives second) in the positive x-axis direction. This allows for a direct comparison between the pump-probe and probe-pump signals.

Concentrating first on the parent ion, we can see that both the pump-probe and probe-pump plots are very similar to one another, showing similar decay rates and similar baseline levels at longer decays (i.e., 500+ fs). The signal of the 69 amu fragment ion, however, does not show this degeneracy. In the case where the pump beam is either the same as, or weaker than, the probe beam (i.e., pump ≤ 3.0 nJ), the pump-probe and probe-pump plots show similar time-dependent features. However, when the pump beam is stronger than the probe beam (i.e., pump > 3.0 nJ), an offset between the two plots is observed, most notable at longer delay times (i.e., 500+ fs). One can see that when using these higher pump powers, the probe-pump signal is stronger than that of the pump-probe signal.

For the remaining fragments ions, a signal above the background is only observed when the pump pulse energy was 6 nJ or above, in addition to the 3 nJ probe pulse. This is in line with what was noted in Section 2.3, where a signal above the background was only observed when the combined energy of the pump and probe pulses was ~9 nJ. Nonetheless, no significant difference between the pump-probe and probe-pump signals was observed for these fragments in these experiments.

## 3. Discussion

In this section, we will first provide an analysis of the VUV PEPICO results alongside the results from the UV/UV Power Series. This will allow us to obtain a general understanding of which VUV photon energies and UV photon orders (at 266 nm) certain ions appear. After this, we will dedicate a section to attributing the fragment ions to their respective chemical formulae, discussing what these fragments mean for the interpretation of 2TU dissociation and how the results compare to past studies. Finally, we will take an in-depth look at the VUV and UV/UV data for each ion, and its role in the interpretation of the ultrafast relaxation dynamics of 2TU.

### 3.1. General Analysis of VUV PEPICO Results and UV/UV Power Series

A number of similarities can be seen between the “Summed” ms-TPES plot from the VUV PEPICO data seen in Figure 2b and the previously published photo-electron spectra of 2TU from Katritzky et al. [48]. The most notable of these are the two distinct bands centred around hv = 9 and 10.5 eV. According to theoretical work by Ruckenbauer et al. [25], the hv = 9 eV band arises due to the ionisation of the n and π orbitals of the sulphur atom, whilst the hv = 10.5 eV band originates from the ionisation of the n and π orbitals of the oxygen atom [25].

For ionisation energies of hv ≥ 11.0 eV, we see that the ms-TPES of the parent ion tends to zero and the fragment ions start to appear. At the same time, the “Summed” ms-TPES signal no longer corresponds as well with photo-electron spectra presented by Katritzky et al. [48] This is likely due to our spectrum being a threshold-electron correlated ion spectrum. The “summed” ms-TPES signal is only representative of the select masses of interest and not of all possible fragments.

The appearance of fragments at these higher photon energies suggests that the ejected electrons are critical to the bonding structure of the pyrimidine ring. From Table 1, one can note that all of the common fragment ions that appear for both 2TU and uracil have a higher appearance energy when originating from 2TU; this is despite 2TU having a lower ionisation energy than uracil (8.73 vs. 9.15 eV) [37]. This may be an indication that the pyrimidine ring is more stable for 2TU, possibly due to the lower electronegativity of the sulphur atom allowing for more electron density to be distributed over the ring. In addition, the appearance of these fragments at these energies is in good agreement with the theoretical work of Ruckenbauer et al. which predicted that ionisation of orbitals located on the pyrimidine ring would begin slightly above 11 eV [25]. In the same theoretical work, a strong ionisation peak between 12–13 eV is predicted and is attributed to the D_4_ ionisation state. This prediction bares resemblance to the large peak seen at ~13 eV in the summed ms-TPES data, with the 69 amu fragment ion being the main contributor to this peak. It is therefore possible that the D_4_ ionisation primarily leads to the production of the 69 amu fragment ion.

Focusing on the UV/UV power series of Section 2.3, we saw that the parent ion was most likely created through a two-photon process when using 266 nm light. This photon order will supply ~9.32 eV to the system, which agrees well with the VUV data, as it falls within ~8.5–11 eV appearance window for the parent ion. This also suggests that the parent ion signal observed in the UV/UV study is due to ionisation of the *n* and *π* orbitals located at the sulphur atom of 2TU [25]. With respect to the fragment ions, we determined that these were all likely created through a three-photon order process or higher (hv ≥ 13.98 eV). This, again, agrees well with the VUV data, as fragments would only appear at energies above 11 eV, and hence a two-photon process is not sufficient.

With these results in mind, one would expect that if one were to use a “low” laser power in the UV/UV experiments, where there are enough photons present for a two-photon order process but high orders are negligible, one would only observe the parent ion. In the lower half of Figure 2c, we present a mass spectrum collected in one of these such “low-power UV/UV” experiments (pink line; 1.5 nJ used between the pump and probe pulses, overlapped at time zero), in which one can see that only the parent ion is observed. In the same plot, we directly compare this to the mass spectra observed in the VUV experiment using photons with an energy of 9.32 ± 0.1 eV (i.e., two photons of 266 nm light, blue line). Once again, only the parent ion is observed, and the two spectra are very similar, despite two different techniques being used.

Similarly, in the upper half of Figure 2c, we compare a high-power UV/UV experiment (black, 13 nJ between the pump and probe pulses, overlapped at time zero) to a mass spectrum from the VUV for photons with energies of 13.98 ± 0.1 eV (red, equivalent to three photons of 266 nm light). In these spectra, we can now see a multitude of fragments have appeared, including signals at 18 and 32 amu, which appear only in the VUV spectrum. These additional peaks most likely come from trace amounts of water vapour (mass = 18 amu, ionisation energy = 12.6 eV [49]) and oxygen gas (mass = 32 amu, ionisation energy = 12.1 eV [50]). Despite these contaminations, the remaining peak positions between the two spectra line up well with one another, further indicating that similar fragment ions are produced when a similar amount of energy is deposited into the system.

Whilst similar fragments are observed in both experiments, differences in the intensities of the ion peaks are apparent, likely due to the differences in the selectivity rules between the two experiments for observing ions. As noted previously, ions in the VUV PEPICO experiment are selectively observed in coincidence with near zero-energy threshold electrons, whilst in the UV/UV experiments no such selectivity is present. This is most exemplified by the parent ion, which appears as an oversaturated signal in the high-power UV/UV experiment, but not at all in the VUV experiment for 13.98 eV photons. The ms-TPES plot seen in Figure 2b shows that the parent ion only appears for photons in the ~8.5–11 eV range, while no threshold photoelectrons are connected to the parent ion at higher photon energies. However, higher electron kinetic energies, not included in the measurement, might very well be correlated to the parent ion. For the UV/UV experiments, the inhomogeneity of the laser focus plays an important role in the observation of the parent. The inner part of the beam is intense and favours three-photon (or higher) processes leading to reduced parent intensity due to reduced dipole matrix elements. The outer part of the beam is of lower intensity and is therefore more suited for inducing two-photon processes. As the outer area of the beam is larger than the inner area, this leads to a significant parent ion signal still being observed.

### 3.2. Fragment Assignment and Discussion of Appearance

The assignment of the major fragments observed in this work has been achieved through comparisons to past dissociation studies on uracil [1,2,35,37,38,51,52,53,54,55,56], as well as studies of 2TU [2,35,36,57]. A summary of the assignments has been included in Table 1. As a formality for this section, we attribute the 128 amu ion to the parent ion, C_4_N_2_H_4_SO^+^.

We attribute the appearance of the 96 amu fragment to the direct loss of the sulphur atom (i.e., 128 − 32 = 96 amu), whilst the 95 amu fragment is most likely due to the combined loss of the sulphur atom and an additional hydrogen atom (i.e., 128 − 32 − 1 = 95 amu). Uleanya et al. also attributed the loss of the sulphur atom to a similar mass fragment in their protonated 2TU studies [36]. Additionally, equivalent ions have been observed in uracil experiments after a loss of oxygen (and hydrogen) [37,53]. We, therefore, attribute the 96 and 95 amu fragment ions of 2TU to the C_4_N_2_H_4_O^+^ and C_4_N_2_H_3_O^+^, respectively.

Whilst for uracil it is difficult to determine if it is the C_2_=O or C_4_=O bond that is broken to produce the 95/96 amu fragment ion [37], no such problem exists for 2TU due to the quasi-isotopic nature between the O and the S atoms. It is clear that the 96 amu fragment ion (and most likely the 95 amu fragment ion) appear due to the breaking of the C_2_=S bond in 2TU [36]. Curiously, in the electron impact studies of the thiouracils by Hecht et al. [35], no peak was observed at 112 amu for 4-thiouracil, which would have indicated a C_2_=O break. Instead, a peak at 95 amu was once again observed, indicating a loss of the sulphur (and hydrogen) from 4-thiouracil. This suggests that the ejection of this unit is more dependent on what type of atoms are attached to the pyrimidine ring, rather than where the atoms are positioned on the ring.

Generally, when the difference between fragments is a single hydrogen atom, the lighter mass has a higher appearance energy [37,38,58]. The fact that the 95 amu fragment ion of 2TU shows a lower appearance energy than the 96 amu ion may therefore suggest that there are novel processes involved in the ejection of the S and H atoms to create the 95 amu. Large vibrational/bending angles in the molecule which bring the sulphur and hydrogen atoms closer together could be one option [39]. Additionally, a migration of one of the atoms on a cationic state before the pair are ejected could be another option [59]. Interestingly, in the photochemistry studies of protonated 2TU by Uleanya et al., where the extra proton sits on the sulphur atom, it was observed that a loss of both the sulphur and hydrogen atoms lead to their most abundant fragment ion signal (their 96 amu = 128 + 1 − (28 + 1)) [36]. With reference to previous studies from Giuliani et al. [60] we are certain that at temperatures employed in our experiment, only the oxo-form of 2TU will appear and hence no tautomers with hydrogen directly bonded to sulphur are present in our studies, furthering the idea that dynamics after ionisation may be at play.

The ejection of the stable (H)NCO fragment is commonly observed in canonical nucleobases [37,46,51,53,55,56,61], and in the case of uracil, it leaves behind a 69 amu fragment ion, which is almost universally attributed to C_3_NH_3_O^+^ [35,37,38,51,52,53,55,56]. It is thought that the ejection of (H)NCO occurs through the breaking of the N_1_C_2_ and N_3_C_4_ bonds of the pyrimidine ring [55]. Like other studies on 2TU [2,35,36], we attribute the 69 amu fragment ion to C_3_NH_3_O^+^, appearing due to the loss of (H)NCS rather than (H)NCO. This suggests that the fragmentation observed here, is more dependent on the nature of the pyrimidine ring, than on the position/substitution of the O/S atoms.

Experiments on deuterated uracil by Ryszka et al. strongly suggest that the 42 amu fragment ion of uracil is C_2_H_2_O^+^ [38], and has been similarly attributed in other studies [2,35,37,53,54,55,56]. This additional evidence from deuterated studies reduces the likelihood that the 42 amu fragment is the less often attributed NCO^+^ ion [52]. Due to the similarities between the molecules, we attribute the 42 amu fragment ion of 2TU to C_2_H_2_O^+^.

Table 1 highlights that the appearance energy for the 41 amu fragment ion is lower than the 42 amu fragment ion. As already noted with the 95 and 96 amu fragment ions, this feature suggests that the difference between the 41 and 42 amu fragment ions is unlikely to be a simple hydrogen loss after 42 amu fragment formation. With this in mind, and after consulting a number of uracil studies [37,38,53,54,55,56], we attribute the 41 amu, at least in its majority, to C_2_NH_3_^+^.

The deuterated-uracil studies of Ryszka et al. suggest that the 28 amu fragment ion is HCNH^+^ [38], which has been backed by several other studies [2,37,53,54]. We give the same attribution to the 28 amu fragment seen from 2TU.

In the case of the 42, 41, and 28 amu fragments, there is the possibility that they could be attributed to other atomic compositions, namely, NCO^+^, HCCO^+^, and CO^+^, respectively, and have been conducted as such in past works on uracil [52,53]. However, these attributions are very much in the minority for uracil and have been superseded by stronger arguments in our own reasoning and the other studies referenced above. We expect the attributions for fragments of 2TU detailed above to be the dominant fragments for each ion observed, and any attributions that may come from other fragments will not significantly contribute to the overall signal observed.

### 3.3. Parent Ion UV/UV Discussions

In Section 2.3, we showed that the parent ion was two-photon dependent at all pump-probe time delays. Combining this with the results seen in Section 2.5, where the same signal was observed in the pump-probe and probe-pump measurements despite using unequal laser intensities in each beam solidifies the idea the time-dependent signal for the parent ion presented in Section 2.4 is the result of a [1 + 1′] resonance-enhanced multiphoton ionisation (REMPI) process.

From the fits of parent-ion pump-probe data presented in Section 2.4, we determined that the observed signal tended to the two-photon auto-correlation signal (430 fs). After reviewing the literature, this is what one would expect when exciting to the S_2_ (^1^π_S_π_6_^*^) state [27], as this has been shown to decay in part directly to the ground state within 200 fs [34], and in part to the S_1_ with a sub-100 fs lifetime [2,24,26,29,31]. For the latter references, the claim is that the S_1_ relaxation is dominant, yet a direct S_2_ → S_0_ decay would explain how the signal returns efficiently to the background level. Additionally, the fact that the expected lifetime of this path is significantly shorter than our 300 fs pulses, would explain why the observed signal tends to be that of the laser auto-correlation signal. Similarly, the S_2_ → S_1_ decay is also expected to be significantly shorter than the duration of our laser pulses, hence why the signal would also tend to the auto-correlation signal for this path. Yet, when the S_2_ → S_1_ path is chosen, one would also expect to see signals relating to the longer-lived S_1_ → Triplet manifold and Triplet → Ground decays—except this is not observed for the parent ion. However, potential energy surface calculations and experimental photoelectron studies by the Ullrich group show that as the excited system propagates towards the S_2_ minima there is a rapid increase in the ionisation energy of the molecule to around ~9.7 eV, and it is not expected to drop back down below this as it passes through the S_1_ and tertiary states [24,29,31]. This means that the energy gap between the initial excitation and the ionisation states is now larger than what a single photon of 266 nm light can provide, and hence it is no longer possible to produce the parent ion through a [1 + 1′] REMPI processes. Other pump-probe ion mass spectroscopy experiments have been similarly insensitive to longer decays, such as those of Ghafur et al. investigating the triplet decay of uracil [2]. It is unfortunately not possible to probe via the parent ion through a [1 + 2′] photon process, as results from Section 2.2, Section 2.3, and Section 2.5 show that this will favour the production of fragment ions instead. All of this explains why we are only able to observe a signal that tends towards the auto-correlation signal for the parent ion. We do, however, note that other experiments have followed the dynamics of the parent ion beyond the S_2_ decay using multi-photon probes with low-energy photons [2], as well as single-photon probe studies with probe energies above ~5.0 eV [24]; this was not possible in our experiment.

### 3.4. Fragment Ion UV/UV Photon-Order Processes

In Section 2.3, we showed that all of the fragment ions are produced in a three-photon process when using 266 nm light at all pump-probe delays, except for the 28 amu ion, which is created in a three-photon process at time zero and proceeds as a four-photon process at all other delays. In Section 2.5, a stronger signal is observed for the 69 amu fragment ion when a weaker pulse is used to pump and a stronger pulse is used to probe. These points together suggest that the 69 amu fragment ion is produced via a [1 + 2′] REMPI process, and, importantly, that the dynamics probed relate to dynamics on the excited neutral system. A reduction of the 69 amu fragment ion signal when a more intense laser is used to excite the system, may also be indicative of rapid relaxations taking place in the cationic states, which prevent further photon absorption at 266 nm, of which similar conclusions have been drawn for the cationic states of uracil in the past [62].

Unfortunately, similar conclusions for the 41, 42, and 95 amu fragment ions cannot be drawn due to conflicting results. In the first instances, the results presented in the Supplementary Data for these fragment ions suggest that a signal is only observed when the pump has a larger power than the probe. From this, we *could* draw the conclusion that the observed signal for these ions originates from a [2 + 1′] REMPI process. However, counter to this, is that in the same plots, we observe no difference between the pump-probe and probe-pump signals. If it were a [2 + 1′] REMPI process, we would expect to see little to no signal when the weaker probe beam comes first; yet no difference is observed. Because of this dichotomy, we cannot explicitly state if the signals observed for these ions relate to a [2 + 1′] or a [1 + 2′] REMPI process, and therefore subsequently if they relate to dynamics on the excited neutral system or the ion. An experiment that could potentially clear this issue, but was not possible to perform here, would be to perform pump-probe experiments in which the pump and probe beams possess different photon energies: one with pulse centred at 266 nm, and the other with a higher photon energy, tuned to allow for ionisation of certain states after excitation. Similarly, in the case of the 28 amu fragment, the results are presented in the Supplementary Data relating to Section 2.5. do not allow us to conclude if the dynamics relate to a [1 + 3′], a [2 + 2′], or a [3 + 1′] REMPI process outside of the pump-probe overlap. Despite these shortcomings, we will continue to draw conclusions from the results for these fragment ions as best as we can.

One of the most interesting observations from Section 2.3 was the change in the photon-order from 3 to 4 from the 28 amu fragment ion when moving away from the pump-probe temporal overlap. This three-photon process at time zero agrees well with the appearance energy of the 28 amu fragment determined in Section 2.2 of 14.04 ± 0.02 eV, which is around the same energy as what is provided by three photons of 266 nm light, 13.98 ± 0.05 eV. The increase in the photon order as we move away from time zero, likely indicates that within the first 100 fs after molecular excitation the energy gap between the pumped state and the ionisation state needed to produce the 28 amu fragment widens along the path of the vibrational wavepacket. This is similar to our argument for the parent ion, as seen in past experiments for 2TU [29] and uracil [4,56]. Unfortunately, due to the unclear results of Section 2.5, we cannot with certainty state if the widening of this gap is occurring between the neutral excited system and the ionisation states, or between the different cationic states.

### 3.5. Fragment Ion UV/UV Time-Dependent Signals

We will start this section with a discussion of the time-dependent signal of the 69 amu fragment ion, due to its clear [1 + 2′] REMPI nature. For this ion, we found that τ_1_ tended towards zero, ultimately matching the shape of the auto-correlation signal. We attribute this to an ultrafast decay that is significantly shorter than the duration of the auto-correlation of the laser, and hence, refers to τ_1_ as a sub-370 fs feature (based on the width of the three-photon auto-correlation signal). This is similar to what was observed with the parent ion, and hence we attribute τ_1_ to either the direct S_2_ → S_0_ decay or the S_2_ → S_1_ decay.

The τ_2_ decay of the 69 amu ion was found to have a duration of 400 ± 300 fs. This agrees well with past studies that suggest the S_1_ → Triplet-manifold decay should have a lifetime in the region of 200-800 fs when photons in the range of 240–300 nm are used to excite the system [2,24,26,29,31].

In the past, the lifetime of the lowest-energy triplet state of 2TU returning to the electronic ground state has been measured to be between ~50 ps to over 200 ps, as noted in Ref. [31]. This, however, was shown to be dependent on the excitation energy used, and corresponds to ~90 ps when ~266 nm light is used—this includes the parent-focussed results of the similar photo-ion probing experiments of Ghafur et al. [2]. We note that the τ_3_ decay of the 69 amu fragment ion was fitted to 310 ± 30 ps, which is significantly longer than the reported parent ion decay. With this in mind, we highlight the strong-field ionisation experiments of uracil by Kotur et al., where a 2.2 ps decay was observed for the parent ion, but a range of decay from 1.5 to 3.5 ps was observed for the fragments [4]. Kotur et al. suggested that the spread may be an indication of a delocalisation of the wavepacket somewhere in the relaxation process, and leads to multiple decay paths. Those could be observed at different points in the potential energy landscape by different probe mechanisms as different time constants. It is possible that a similar process is also observed here for 2TU.

In Table 3, we can see that the 28, 41, 69, and 95 fragment ions all share similarly fitted decay constants. All of the τ_1_ constants tend towards zero, all values for τ_2_ fall in the region of 300–400 fs, and all of the τ_3_ constants fall in the range of 220 to 400 ps. Due to the similar decay constants between all of the ions, we believe that this is indicative that the decay constants are representative of the same decay mechanisms. That is to say, we believe that all of the fragment ions discussed here follow the same decay process as that ascribed to the 69 amu fragment ion; i.e., τ_1_ is the S_2_ → S_0_/S_2_ → S_1_ decay, τ_2_ is the S_1_ → Triplet-manifold decay, and τ_3_ is the Triplet Manifold → ground state decay. This, therefore, implies that the signals observed for these fragment ions are representative primarily of dynamics occurring on the neutrally-excited system and not the ion. If a significant contribution were due to dynamics initiated on the ionic species (i.e., a [2 + 1′] REMPI process) we would expect different, if not additional, decays to those observed here and in the literature. However, as this conclusion is based on coincidence, we cannot explicitly confirm this due to ambiguities noted in Section 2.5 and Section 3.4.

Finally, the 42 amu ion appears to be somewhat of an outlier for all of the ions we discuss here as no unique τ_2_ decay could be fitted. In Table 3, we can see that for all of the other fragment ions, the value of τ_2_ varies slightly, yet are on the same order of magnitude as the three-photon autocorrelation signal (370 fs). We, therefore, suspect that the τ_2_ decay is still present for the 42 amu ion, however, it may be sufficiently shorter than the other τ_2_ values so as to be obscured by the relatively wide auto-correlation signal. Therefore, formally, we attribute the τ_1_ decay of the 42 amu ion to an overlap of the S_2_ → S_0_/S_2_ → S_1_ decay and the S_1_ → Triplet-manifold decay, whilst the τ_3_ decay for the 42 amu ion still represents the Triplet Manifold → ground state decay.

## 4. Materials and Methods

### 4.1. UV/UV Pump-Probe Experiments

In this section, we provide a general overview of the setup used in the UV/UV pump-probe experiments. As noted previously, specific parameters of each of the experiments discussed in the results section are summarised at the beginning of each respective subsection.

Figure 6 shows a simplified version of the experimental set-up used in the UV/UV studies, and is similar to the set-up used in previously published experiments [47]. In this, a 1 W, 1 kHz, 35 fs, 800 nm beam from a Ti:Sapphire laser is directed through a third harmonic generation (THG) set-up. The UV pulse energy is controlled by detuning the input beam polarisation with respect to the THG set-up via the rotation of a λ/2 plate situated before the THG set-up. After generation, the filtered third-harmonic light passes through a Michelson interferometer to produce collinear pump and probe beams, of which the delay between the two beams is controlled via a motorised translation stage. With the interferometer described so far, pump and probe beams of equal powers will be produced; this is the setup used in Section 2.1, Section 2.3, and Section 2.4. For results relating to those seen in Section 2.5, variable ND filters were inserted into the individual arms of the Michelson interferometer, allowing for the production of pump and probe beams with individually controlled pulse energies. After the interferometer, the beams are directed towards the vacuum apparatus via a number of reflectors and focussing optics. The spot size of the UV laser beams was measured to be approximately 30 μm in diameter. A power meter positioned directly at an exit viewport of the apparatus (after the laser is focussed in the chamber) was used to measure the power of the laser for each data point collected.

Inside the vacuum apparatus, the sample capillary oven [63] is positioned using a three-axis linear manipulator over the path of the pump and probe beams, with the tip of the delivery needle sitting ~1–2 mm above the laser focus. As molecules leave the oven, they are ionized by the laser pulses and detected via a Time-of-Flight (ToF) mass spectrometer based on the Wiley-McLaren design [64] using micro-channel-plate detectors. The ToF traces are read out using an oscilloscope. All data from the oscilloscope are transferred to a computer, where all traces are recorded. The repeller and extractor plates of the Wiley-McLaren electron-optics are held at +900 and −900 V, respectively, and are situated 30 mm apart. The 180 mm long flight tube starts 10 mm behind the extractor and is set to −1600 V to further accelerate the ions. Voltages on each component were slightly tuned while keeping differences constant to optimize resolution. The achievable mass resolution of the spectrometer assembly was about 150.

The 2-thiouracil was obtained from Sigma-Aldrich (purity ≥ 99%) and used without further refinement. The capillary oven was heated to 190 °C, which is below the temperature used in other experiments, so we are confident that we are not at the decomposition threshold [24,28,31]. The nozzle of the oven is heated to 200 °C to prevent sample delivery blockages. This produced a typical sample density of 1 × 10^12^ molecules/cm^3^ at the point where laser beams cross with the gas flow [63]. Due to the high temperatures and effusive nature of the source, we expect clustering to be negligible and no higher-order clusters were observed in the mass spectra.

The pulse duration of the UV laser was obtained by recording the time-resolved mass spectra of different gases backfilled into the apparatus through a needle valve, whose output was situated near the focus of the pump and probe beams. These measurements were performed with the variable ND filters seen in Figure 6 removed. The gases used were 1,4-dioxane and xenon, which are ionised via a two- and three-photon non-resonant process, respectively, when using 266 nm photons [65,66]. The time-dependent parent ion signals for 1,4-dioxane and xenon were fitted to a Gaussian curve with full-width half-maxima (FWHM) of 430 and 370 fs, respectively. Both tests suggest a pulse duration of 300 fs. This is in line with what is expected when a 35-fs-long pulse, centred at 266 nm, passes through ~20 mm of UVFS glass (arising from the beamsplitter, telescope, and viewport), which accumulates a total dispersion of ~3800 fs^2^. Through fits of data seen in Section 3.5, we estimate that the pulses are stretched to ~500 fs with the aforementioned ND filters in place. This agrees with dispersion calculations, which suggest that an additional ~1900 fs^2^ dispersion is accumulated on top of the value stated above, as the laser makes a double pass through the ~5 mm thick UVFS ND filters. The stretching of the pulse will not affect the observed dynamics for the time constants discussed here.

Data were collected with both the pump and probe beams entering the chamber in a “laser-on/laser-off” fashion. For each laser-on data point collected, a laser-off data point was collected immediately afterwards, before moving to the next data point. This allows for improved background correction. In addition, runs that relate to pump-only and probe-only signals were also collected, allowing for “pump-/probe-only” Vs “pump-probe” signals to be compared where necessary. Data were collected in a series of “loops”. For each loop, data for the full combination of pump-probe delays and laser powers of interest were collected in a randomized order before the process was repeated for the next loop. This method helps to reduce the effect of slow drifts, whilst the combination of small instabilities in the interferometer and the multiple samples taken help to reduce interference effects between the pump and probe beams around time zero. In the power series of Section 2.3, a total of 24,000 laser shots per data point were collected over the course of 8 loops. In the pump-probe scans of Section 2.4, data representing 50,000 laser shots per data point were collected over 16 Loops for the “low-power” scan, over the delay range of −1 → 1 ps. In the “high-power” scan performed over −3 → 15 ps, 250,000 laser shots per data point were collected over 50 Loops, whilst in the “high-power” scan over the range of 0 → 100 ps, 60,000 laser shots per data point were collected over 12 Loops. Finally, in the non-equal pump-probe experiments of Section 2.5, data was collected over 10 loops with a total of 14,000 laser shots being collected per data point.

The laser powers used depended on the experiment being performed, and exact values are detailed as appropriate in the following sections. However, in general, experimental pulse energies were measured to be between 0.5–15 (±5%) nJ per beam, providing a beam intensity in the range of 2 × 10^8^–7 × 10^9^ W/cm^2^ per beam at the interaction region.

### 4.2. VUV Dissociative Photoionsation Experiments

The VUV dissociative photoionisation data were collected at the VUV beamline of the SLS. The instrument for our study utilises a double velocity-map-imaging photoelectron photoion coincidence (‘PEPICO’) spectrometer to detect ions produced by a sample coming into contact with a quasi-continuous-wave monochromatized synchrotron light source [67,68,69,70]. Specifically, we selectively observe ions that arrive in coincidence to near zero-kinetic energy electrons (circa. <10 meV), allowing us to study threshold-ionized ions (i.e., threshold photoelectron photoion coincidence spectroscopy). Ultimately, this allows for the determination of the appearance energies of the ions of interest. The same 2TU sample described in the UV/UV experiment was used here, introduced into the path of the VUV light via a filled ampule heated to 150 °C, and directed towards the interaction region via a thin nozzle.

Data were collected in three runs, each over a different range of photon energies, which include: (1) hv = 8 → 12 eV in 10 meV steps, (2) hv = 11 → 15 eV in 20 meV steps, and (3) hv = 12 → 20 eV in 40 meV steps. Whilst data up to 20 eV photon energy was recorded, only data in the hv = 8.55–15 eV range is presented in this paper, as this is sufficient to assist in the analysis of the UV/UV experiments. The data presented in Figure 2 is stitched together from the three runs, explicitly using data from run (1) in the range hv = 8.55 → 11.5 eV, run (2) in the range hv = 11.5 → 14 eV, and run (3) in the range hv = 14 → 15 eV. Switch-over points were chosen to account for both signal strength and the effects of the gas filter of the beamline [70]. The smallest step size of 10 meV approximately matches the bandwidth of the monochromatic beam [67]. This collection method produced a fine scan over the region where ions of interest would start to appear, as well as providing sufficient overlap between each run, allowing for normalisation of signal between runs, and accounting for changes of gas filters in the beamline [70]. For each data point, 180 s worth of data was collected.

## 5. Conclusions

In this work, through the combination of time-resolved UV/UV pump-probe data and VUV PEPICO data, we have been able to gain a deeper understanding of the excited-state dynamics of 2TU and its photo-induced dissociation. We have been able to use experimental evidence from two different techniques to provide reasoning as to why different photon orders are needed for the production of different ions. In the case of the 28 amu fragment ion, we were also able to provide reasoning as to why this photon order changes once a delay is introduced between the pump and probe beams.

Additionally, in fitting the time-dependent signals of the fragment ions, we find that their decay constants match well to the previously established S_2_ → S_1_ → Triplet manifold → ground state decay route for 2TU, with the hint that the exact Triplet manifold → ground state decay constant may be fragment specific.

We have also highlighted and discussed the origin of several fragment ions, such as the difference between the 95 and 96 amu ions, which may in the future prove useful in identifying the differences in the dynamics between 2TU and uracil, as well as 4-thiouracil and 2,4-dithiouracil.

Overall, the results presented here provide information on the higher-lying ionisation states of 2TU, and, to a degree how, they evolve as 2TU decays from the S_2_ state. This information will be useful for future theoretical models of 2TU that concentrate on these states, as well as other (thio)uracils and DNA/RNA nucleobases.

## Figures and Tables

**Figure 1 molecules-28-02354-f001:**
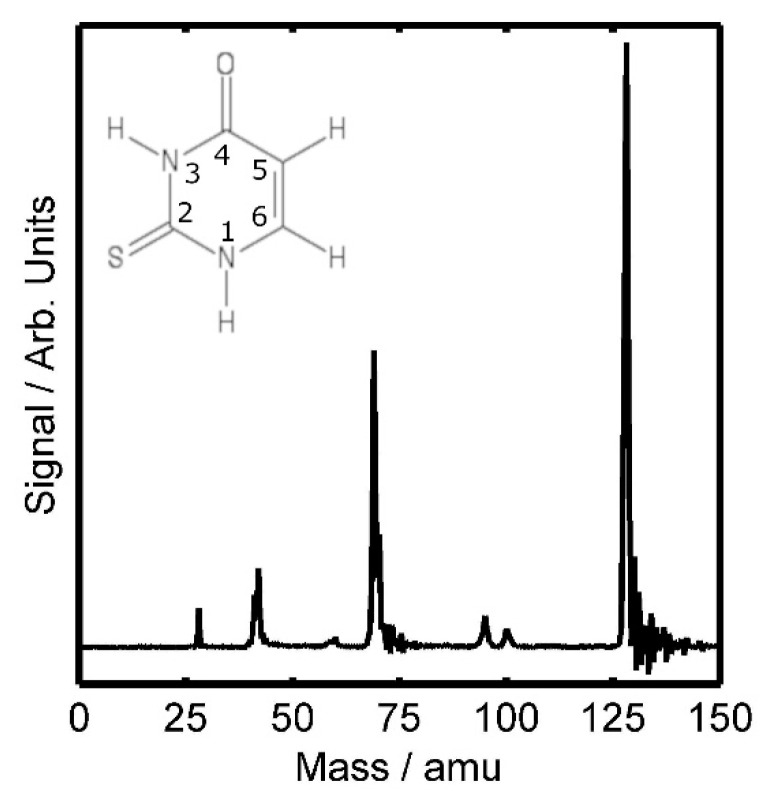
Typical photo-ion mass spectrum of 2TU taken from the UV/UV experiment. Data were recorded at the pump-probe temporal overlap, with a combined pump-probe energy of 13 nJ. The insert shows the numbered 2TU molecule. As we do not have indications of fragment charges higher than one, the x-axis gives the ion mass.

**Figure 2 molecules-28-02354-f002:**
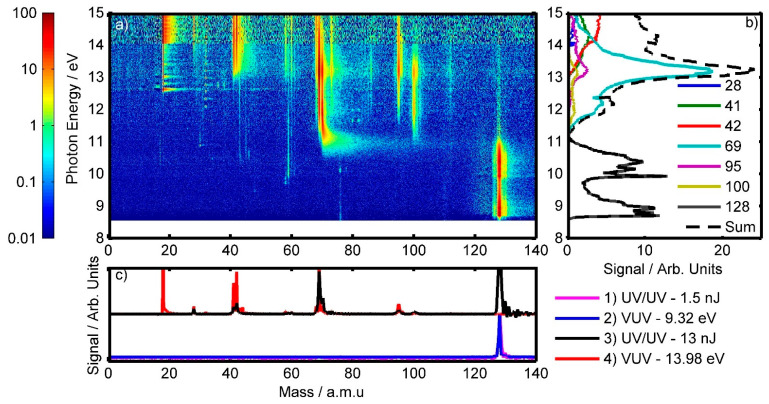
(**a**) Logarithmic false-colour representation of the background-corrected VUV data collected at SLS showing threshold-ionisation-selected ions collected via the PEPICO method for photon energies between 8.55 and 15 eV. (**b**) ms-TPES for select fragments. (**c**) Cuts of the VUV data along the mass axis at energies that correspond to 2-photon (9.32 eV) and 3-photon (13.98 eV) ionisation at 266 nm, overlain with low- and high-power signals from the UV/UV experiment, respectively, obtained at time-zero.

**Figure 3 molecules-28-02354-f003:**
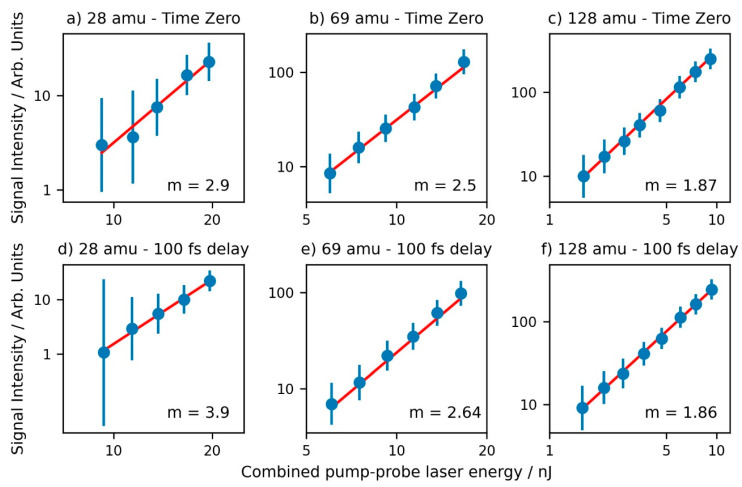
Power series plots (displayed in a “log-log” format) showing how the Signal varies with respect to the combined pump-probe laser energy for the 28 amu fragment ion (parts **a**,**d**), the 69 amu fragment ion (parts **b**,**e**), and the parent (128 amu) ion (parts **c**,**f**) for the time-zero overlap (parts **a**–**c**) and at 100 fs pump-probe delay (parts **d**–**f**). A linear fit of these data points is shown by the solid red line. The data range fitted was chosen in an area where the signal was above the background noise and below saturation. The value of the fitted gradient, *m*, is shown in the bottom right corner of each plot, its measurement uncertainties are given in Table 2.

**Figure 4 molecules-28-02354-f004:**
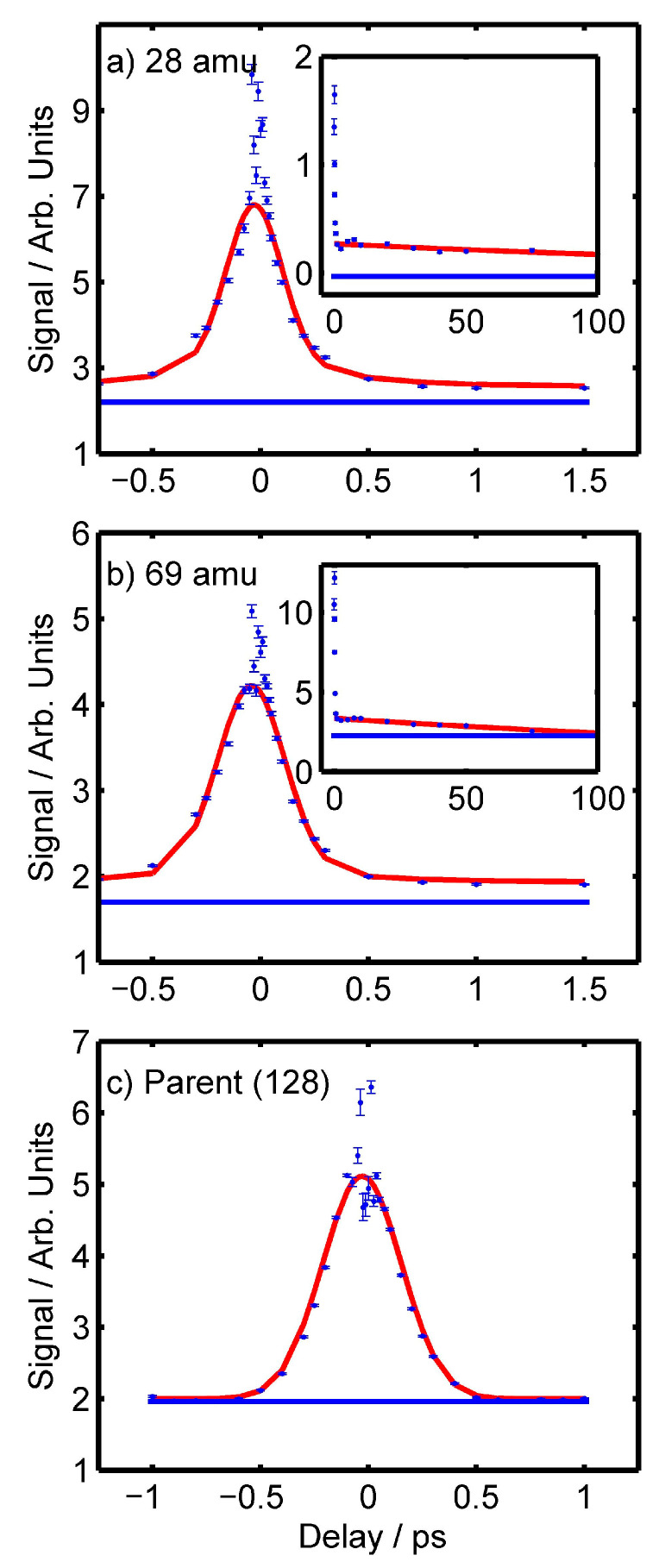
Time-resolved signals for the (**a**) 28 amu (**b**) 69 amu and (**c**) Parent (128 amu) ions. The original data is represented by the blue dots, the fit of this data by the red line, and the background signal by the blue line. In (**a**,**b**) inserts show the respective signal for the fragment ions extending out to 100 ps.

**Figure 5 molecules-28-02354-f005:**
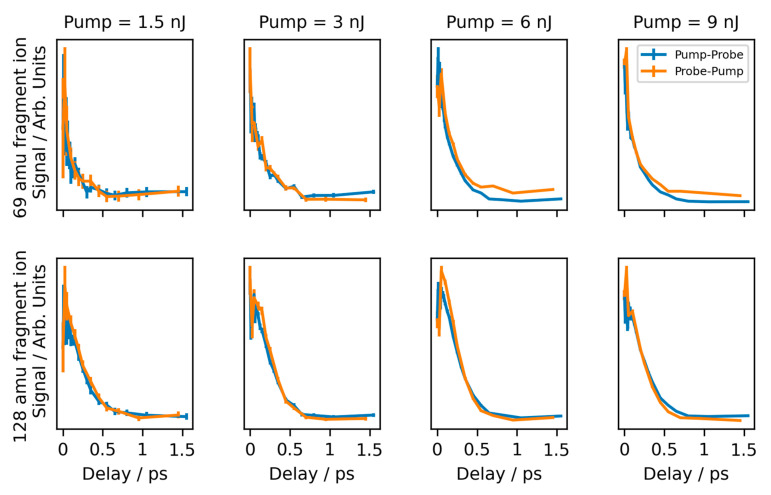
Time-dependent signals observed for the 69 amu fragment ion (**Top**) and the parent (128 amu) ion (**Bottom**) when unequal pump and probe powers are used. The measurement values are given by the symbols and are connected by lines. The uncertainty interval is given by the height of the symbols; however, these are sometimes obscured by the lines. Signals from a range of pump pulse energies were used (from **left** to **right**: 1.5, 3, 6, and 9 nJ) in conjunction with the probe pulse, which was fixed at 3 nJ. In each image, two plots are shown which dictate whether the pump pulse arrives first (pump-probe, blue) or if the probe pulse arrives first (probe-pump, orange).

**Figure 6 molecules-28-02354-f006:**
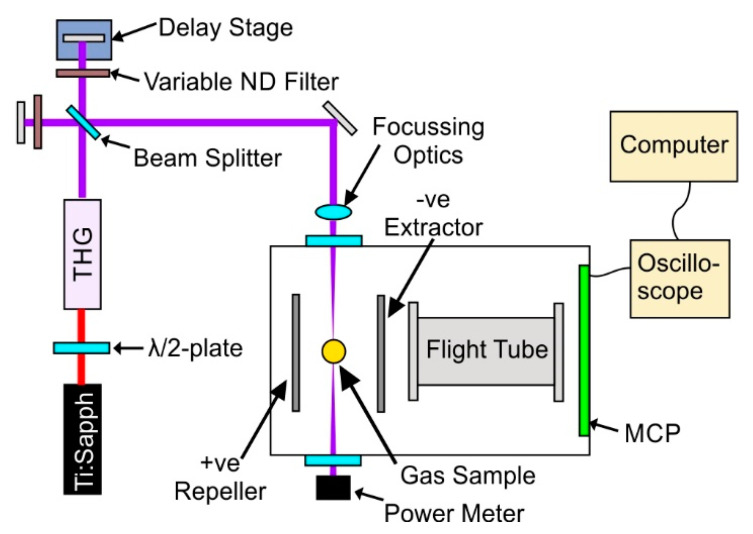
Simplified sketch of the experimental set-up, showing a Michelson interferometer creating pump and probe pulses with controlled delay from the third-harmonic (266 nm) of a Ti:Sapphire laser. Variable ND filters sit in each arm of the interferometer, allowing for pump and probe beams with differing powers to be utilised. After the interferometer, the laser beams are directed into the ion mass spectrometer, where they cross with an effusive beam of 2TU (produced from a heated capillary oven [63]) to induce ionisation. These ions are accelerated via an electric potential across extraction plates towards an MCP detector through a flight tube, where their m/q-dependent arrival time is observed on an oscilloscope and recorded on a computer.

**Table 1 molecules-28-02354-t001:** Attribution of the fragment ions of 2TU to chemical formulae, along with the respective appearance energies observed in the VUV photofragmentation studies, fitted using the Asher model [40]. Comparisons are also made to similar fragments observed in the uracil work of Jochims et al. [37].

Mass/Amu	Attributed Formula	Appearance Energies/eV	Appearance Energy in Uracil/eV [37]
28	HCNH^+^	14.04 ± 0.02	13.75 ± 0.05
41	C_2_NH_3_^+^	13.47 ± 0.01	12.95 ± 0.05
42	C_2_H_2_O^+^	13.83 ± 0.01	13.25 ± 0.05
69	C_3_NH_3_O^+^	11.90 ± 0.01	10.95 ± 0.05
95	C_4_N_2_H_3_O^+^	12.65 ± 0.02	Observed, but not given
96	C_4_N_2_H_4_O^+^	13.59 ± 0.01	Observed, but not given
100	C_3_N_2_H_4_S^+^	11.95 ± 0.01	Not observed
128 (Parent)	C_4_N_2_H_4_SO^+^	8.73 ± 0.01	9.15 ± 0.03

**Table 2 molecules-28-02354-t002:** Fitted slope values for Log_10_(Signal) Vs Log_10_(Laser Energy) plots for the fragments of interest observed in the UV/UV experiment at various pump-probe delay times.

Delay (ps)	28 amu	41 amu	42 amu	69 amu	95 amu	Parent
0.0	2.9 ± 0.5	2.68 ± 0.15	2.72 ± 0.14	2.5 ± 0.3	2.6 ± 0.2	1.87 ± 0.06
0.1	3.9 ± 0.3	2.87 ± 0.06	2.90 ± 0.10	2.64 ± 0.05	2.74 ± 0.09	1.86 ± 0.04
0.5	3.5 ± 0.7	2.76 ± 0.16	2.9 ± 0.2	2.8 ± 0.4	2.48 ± 0.12	1.75 ± 0.11
1	3.2 ± 1.2	2.8 ± 0.4	2.8 ± 0.3	2.70 ± 0.15	2.7 ± 0.5	1.75 ± 0.05
5	3.6 ± 0.3	2.90 ± 0.08	2.91 ± 0.10	2.71 ± 0.10	2.80 ± 0. 16	1.99 ± 0.06
10	3.6 ± 0.3	2.9 ± 0.2	2.95 ± 0.05	2.71 ± 0.05	2.7 ± 0.3	1.86 ± 0.03

**Table 3 molecules-28-02354-t003:** Fitted decay constants for the parent and fragments of 2TU as observed in the UV/UV time-resolved experiment.

Ion	τ_1_/fs	τ_2_/fs	τ_3_/ps
Parent	Tends to zero	N/A	N/A
28		330 ± 150	220 ± 60
41		400 ± 300	340 ± 60
42		N/A	300 ± 50
69		400 ± 300	310 ± 30
95		340 ± 130	380 ± 160

## Data Availability

Processed, background subtracted data that is presented in the figures of this article can be found at: https://figshare.com/s/3fc2bee30b8fb40171d3 (accessed on 18 October 2022). Larger, raw data files, which are serval GB in size, can be requested from the authors directly.

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
