# Peer review of "Ultrafast Photo-Ion Probing of the Relaxation Dynamics in 2-Thiouracil"

_molecules, 2023, doi:10.3390/molecules28052354_

Round 1

Reviewer 1 Report

This manuscript presents a series of UV and VUV driven ionization measurements of 2-thiouracil.  The authors perform VUV photoion-photoelectron coincidence measurements looking at near zero energy electrons (mass selected threshold ionization measurements) using VUV photons from the Swiss light source and UV pump UV probe time resolved photoion spectroscopy measurements, using time of flight mass spectroscopy to follow the different photoion yields as a function of pump probe delay.  They interpret their measurements to try to develop a better understanding of the UV initiated ultrafast excited state dynamics. 

I think that the manuscript is generally well written and clear (except for the consistent “Error! Referefence source not found” issue).  In particular, I found the introduction and motivation to be quite nice.  The combination of UV and VUV measurements is useful for interpreting the different fragment ion yields.  However, I have a number of comments about the manuscript, most of which are concerned with the UV/UV experiments and the interpretation of their results.  Here are a number of concerns, which I think that the authors should address prior to publication:

1/  I think that one of the most significant problems is the fact that the authors use the same UV pulse for pump and probe.  As figure 5 shows, they don’t see much difference between positive and negative time delays.  Since the fragment ions all involve 3 photon absorption, both [2+1’] and [1+2’] REMPI processes contribute.  For similar signals on both sides of timezero, this means that the authors cannot really distinguish between 2 photon driven dynamics and 1 photon driven dynamics (i.e. absorbing 2 pump photons or 1 pump photon).  Thus it seems to me that it is hard to say much about the dynamics in the excited state of interest because their signal could be dominated by (or at least have nonzero contributions from) cationic dynamics which are launched by two photon absorption to the ground state of the cation (or even neutral states above the ionization threshold which can have comparable oscillator strengths to ionization) and then probed by one photon absorption to a dissociative state of the cation.  I didn’t find the pump probe scans for different pump pulse energies very illuminating since the differences were very small and restricted to long times. 

2/  In addition to the issue discussed in point 1 above, it seems to me that much of the dynamics that the authors are interested in uncovering occur on timescales faster than the impulse response function (IRF) of the apparatus.  To the extent that the pulse duration or IRF is significantly longer than the dynamics, the measurements serve more as a measurement of the pulse duration or IRF than of the molecular dynamics in question.  While the authors fit their data to pull out some picosecond time constants, the signals have already decayed to less than 10% of the yield at t=0 by 1 ps for all pump pulse energies and fragments shown in figure 5.  Also, on a related note, it is not clear what the motivation was for the fitting function used by the authors. Is there a kinetic model which they used, some other model, or was it simply empirically based? 

3/  It is not clear to me why the pump and probe pulses are so long.  With the amplifier output being ~35 fs, I would have imagined that it is possible to produce an IRF of less than 100 fs, even without dispersion compensation using prisms.

4/  The way in which the authors treated the coherent artifact at zero time delay was not very compelling.  If it is related to interference between pump and probe pulses, why is it confined to a small section of the signal near zero time delay?  Perhaps it is due to the pulses being chirped and the coherent artifact is a measure of the coherence time – i.e. the spectrum of the pulses could support a shorter duration (the region over which there is optical interference), but due to dispersion, they become chirped and then do not interfere for longer delays?  Are the delays between the pump and probe arms phaselocked?  How did the authors determine what is coherent artifact and what is “molecular signal” when fitting the results shown in figure 4?  

5/  I think that the extension of the trend line below the region where the fitting was carried out in figure 3 is a little misleading/confusing.

In conclusion, I think that the manuscript contains some interesting results and discussion, but the connection between the measurements and dynamics is a little tenuous and could be strengthened.    

Reviewer 2 Report

The authors investigate the relaxation processes occurring in 2-thiouracil (2-TU) following photoexcitation to the S2 state using degenerate (266 nm) UV pump/probe spectroscopy and tracking the appearance of various fragment ions as a function of pump/probe time delay (ie transients). These time-resolved measurements are complemented by single-photon VUV-induced dissociative photoionization studies that provide appearance energies of these fragment ions. This enables one to correlate the appearance energy (of the fragment ions) and photon order (in pump/probe studies).  

In addition to the parent ion transient, whose appearance is instrument limited, the authors observe three major decay channels for the fragment ions. The first is instrument limited; the second is on the order of a few 100 fs; and the third is between 200-400 ps. With guidance from previous studies, these authors assign the dynamics to internal conversion from S2-S1(instrument limited); intersystem crossing from S1-Tn (few 100 fs); and back ISC to repopulate the electronic ground state, ie Tn-S0.

All in all, this is a nice study, demonstrating the rich information that can be gleaned about the relaxation processes, and hence photostability, of thionated nucleobases. Consequently, I believe this work merits strong consideration for publication in molecules. I would however urge the authors to consider the following points in my ‘minor revision’ request. There is no need for me to see a revised version of this manuscript.

1.     Do the authors see any influence from clustering? I did not see any mention of this (apologies if I missed this) so it would be good to briefly mention (as they did with decomposition) how clustering was mitigated.

2.     I was somewhat surprised that the measured FWHM was around 400 fs given the authors start with a 35 fs fundamental pulse. Granted, dispersion for the 266 nm will be fairly large but I would not have expected the pulse to broaden to that approaching 400 fs (excluding NDFs). Perhaps it has something to do with the l/2 waveplate and sub-optimal phase-matching? Or alternatively using thick optical substrates (beam splitter)?

3.     Issues with referencing of figures. But I suspect this may have arisen when the manuscript was uploaded/compiled.

Round 2

Reviewer 1 Report

I appreciate the response of the authors.  They have addressed some of my concerns, but I am still a little worried about the UV/UV time dependent ion yields.   In particular, the authors conclude based on the results shown in figure 5 that the 69 amu ion yield arises from a 1+2’ REMPI process, and therefore provides information on the excited neutral state dynamics. However, I still have some questions about this analysis. The authors conclude that since the differences between pump-probe and probe-pump results are bigger than the error bars, there are both 2+1’ and 1+2’ dynamics (different from the parent ion, which is just 1+1’).  If I understand correctly, then they further argue that the pump probe results for small pump pulse energies correspond to 1+2’ REMPI dynamics – i.e. pumping to a neutral excited state and then probing to the cation.  But can’t there be both 1+2’ and 2+1’ contributions to both sides of timezero?  I agree that their measurements suggest that there is an asymmetric mix of the two for times greater to or less than zero delay, but what if this mix is 60%/40% (t<0) and 40%/60% (t>0) (or vice versa)?  Is there any way of being a bit more quantitative about this mixture, and how this allows one to assign timescales for the excited neutral state separate from two photon pumped dynamics? If it is not possible to determine exactly what the mixture is, can the authors place some limits on the contributions or timescales?  Also, the blue curves for 1.5 nJ and 3nJ pump show a minimum in the yield around 500 fs followed by an increase for longer time delays, which is also larger than the error bars. This seems to suggest some subtleties in the dynamics or probing.  Do they authors have any ideas for what would lead to this? Finally, two additional minor points: 1/ if the error bars are obscured by the lines, why not make the lines thinner or plot the error bars over the lines such that they are visible? The authors note in their response letter that they have updated the error bars, but they were not clear to me.  2/ It seems a little strange to me to quote three significant digits for an uncertainty in a fitted decay time. Typically, I would expect only one (or two digits if the first digit is a 1) in a quoted uncertainty. 
